# *Rhodotorula mucilaginosa* ZTHY2 Attenuates Cyclophosphamide-Induced Immunosuppression in Mice

**DOI:** 10.3390/ani13213376

**Published:** 2023-10-31

**Authors:** Kai Kang, Xinyi Deng, Weitian Xie, Jinjun Chen, Hongying Lin, Zhibao Chen

**Affiliations:** 1Department of Veterinary Medicine, College of Coastal Agricultural Sciences, Guangdong Ocean University, Zhanjiang 524088, China; kangkai610@126.com (K.K.); dengxinyi1110@163.com (X.D.); xieweitian@163.com (W.X.); chenjj@gdou.edu.cn (J.C.); gdoulinhongying@sina.com (H.L.); 2South China Branch of National Saline-Alkali Tolerant Rice Technology Innovation Center Zhanjiang, Zhanjiang 524088, China

**Keywords:** *Rhodotorula mucilaginosa* (*R. mucilaginosa*), cellular immunity, humoral immunity, cyclophosphamide (CTX), immunosuppression

## Abstract

**Simple Summary:**

*Rhodotorula mucilaginosa* is widely detected in the environment, and it produces broad metabolites such as carotenoid pigments, amino acids, polyunsaturated fatty acids, polysaccharides, and enzymatic substances. We previously isolated a strain of *Rhodotorula mucilaginosa* ZTHY2 from the coastal waters of the South China Sea, which showed probiotic properties. *Rhodotorula mucilaginosa* ZTHY2 promoted the growth of experimental animals, increased the abundance of beneficial bacteria in gut, and enhanced immune function. Given that immunosuppression is common in livestock, environmental stresses, such as overpopulation, food contamination, and chronic diseases, contribute to immunosuppression of livestock associated with declining animal production, failed immunoprophylaxis, and economic losses. Therefore, in this study, cyclophosphamide was used to generate immunosuppressed models to further evaluate the immunomodulatory effects of *Rhodotorula mucilaginosa* ZTHY2.

**Abstract:**

*Rhodotorula mucilaginosa* (*R. mucilaginosa*) can enhance the immune and antioxidant function of the body. However, whether *R. mucilaginosa* has an immunoregulatory effect on cyclophosphamide (CTX)-induced immunosuppressed animals remains to be clarified. In this study, the *R. mucilaginosa* ZTHY2 that we isolated from the coastal waters of the South China Sea previously was prepared in order to investigate its immunoprotective effect on CTX-induced immunosuppression in mice, and the effects were compared to those of *Lactobacillus acidophilus* (LA) (a well-known probiotic). Seventy-two male SPF mice were divided into six groups: The C group (control); IM group (immunosuppressive model group) (+CTX); Rl, Rm, and Rh groups (+CTX+low, medium, and high concentration of *R. mucilaginosa*, respectively); and PC (positive control) group (+CTX+LA). After a 28-day feeding trial, blood samples were taken for biochemical and serum immunological analysis, and the thymus and spleen were collected to analyze the organ index, lymphocyte proliferation and differentiation, and antioxidant capacity. The findings showed that *R. mucilaginosa* ZTHY2 improved the spleen and thymus indices, effectively attenuated immune organ atrophy caused by CTX, and enhanced the proliferation of T and B lymphocytes induced by ConA and LPS. *R. mucilaginosa* ZTHY2 promoted the secretion of cytokines and immunoglobulins and significantly increased the contents of IL-2, IL-4, IL-6, TNF-α, IFN-γ, IgA, IgG, IgM, CD4, CD8, CD19, and CD20 in serum. The proportion of CD4^+^, CD8^+^, CD19^+^, and CD20^+^ lymphocytes in spleen, thymus, and mesenteric lymph nodes were increased. In addition, *R. mucilaginosa* ZTHY2 reduced the reactive oxygen species (ROS) and malondialdehyde (MDA) levels and increased glutathione (GSH), total superoxide dismutase (SOD), and catalase (CAT) levels. Our results indicated that *R. mucilaginosa* ZTHY2 can significantly enhance the immune function of immunosuppressed mice, and improving antioxidant capacity thus attenuates CTX-induced immunosuppression and immune organ atrophy.

## 1. Introduction

*Rhodotorula mucilaginosa* (*R. mucilaginosa*), known as “red yeast”, is widely detected in the sea, fresh water, soil, animals, and plants, etc., and it produces carotenoid pigments, resulting in orange-, pinkish-, or red-colored colonies [1]. *R. mucilaginosa* has been isolated and cultivated for various purposes, such as its bioproducts. *R. mucilaginosa*’s metabolites include amino acids, polyunsaturated fatty acids, polysaccharides, and enzymatic substances; *R. mucilaginosa* is thus an economic and valuable resource for biomedical and agricultural products [2,3]. *R. mucilaginosa* possesses valuable biotechnological features, although there have been a few case reports of fungemia, inflammation, or skin infection in humans, particularly in patients with immunodeficiency, tumors, or chronic disease [4,5,6,7,8]. However, environmentally derived *R. mucilaginosa* has been shown to promote growth in animals. *R. mucilaginosa* has been used as an additive to fodder for increasing the growth and antioxidant capacity of weaned piglets [9] and specific growth rate (SGR) and protein content of juvenile Nile tilapia [10]. An *R. mucilaginosa* solid-state fermentation product promoted the production of laying hens [11]. *R. mucilaginosa* has also been utilized by the food industry to produce enzymes, such as phenylalanine ammonia lyase (PAL) [12]. *R. mucilaginosa* has been shown to enhance immune responses in aquatic arthropods and vertebrates [10,13]; however, whether *R. mucilaginosa* helps enhance immunity of terrestrial animals remains to be determined.

The immune response of terrestrial animals depends on the innate and adaptive immune systems. The innate immune system provides the first line of defense against “non-self” factors, and the adaptive immune system consists of humoral and cell-mediated immunity to control infections [14,15]. Lymphocytes and macrophages produce and secrete inflammatory factors, such as tumor necrosis factor-α (TNF-α), interferon-(IFN)-γ, interleukin (IL)-2, IL-4, and IL-6. Both lymphocytes and macrophages play important roles in the function and homeostasis of animal immune systems. Altered homeostasis of the immune system contributes to immunosuppression associated with pathophysiological and environmental stresses [16,17]. Environmental stresses, such as overpopulation, food contamination, and chronic diseases, contribute to the immunosuppression of livestock associated with declining animal production, failed immunoprophylaxis, and economic losses [18,19,20,21]. Although natural products have been considered to help maintain the immunity of livestock [22,23,24], natural products effective in modulating immunosuppression remain to be determined.

Cyclophosphamide (CTX) is an antineoplastic drug for lymphoma and an immunosuppressant to treat autoimmune diseases, such as severe systemic lupus erythematosus [25,26]. CTX is also commonly used as a cancer treatment in companion animals [27,28,29,30], and the administration of CTX often leads to adverse effects in treated companion animals such as lethargy, moderate alopecia, vomiting, anorexia, anemia, and hematuria [31,32,33]. In addition, CTX is routinely used to induce immunosuppression in animal models associated with bone-marrow suppression, atrophy of the spleen and thymus, an imbalance of blood cells, and suppression of cytokines, such as IL-2, IL-4, TNF-α, IFN-γ, IgA, IgG, and IgM [34,35,36,37]. CTX has also been shown to suppress antioxidant superoxide dismutase (SOD) and catalase (CAT). SOD is a scavenger of oxygen free radicals, and CAT decomposes hydrogen peroxide into O_2_ and H_2_O as well as increases the lipid peroxidation product malondialdehyde (MDA), an index for oxidative damage [38,39]. Accordingly, CTX acts an optimal agent for studying immunosuppression and oxidation.

In our previous studies, we reported a novel strain of *R. mucilaginosa* ZTHY2 isolated from the sea of Southwest China [40], and we have successfully extracted astaxanthin from it [41]. Feeding animals with *R. mucilaginosa* ZTHY2 increased the thymus and spleen indices, delayed hypersensitivity, and increased IgG, IgA, IL-2, TNF-α, and INF-γ in serum [42]. Accordingly, *R. mucilaginosa* ZTHY2 appeared to help enhance the immune system. In this study, in order to investigate the effect of *R. Mucilaginosa* ZTHY2 on immunoprotective function and antioxidant capacity in CTX-induced immunosuppressed mice, the mechanisms were addressed by analyzing the immune organ index, blood biochemical factors, immunoglobulins, inflammatory factors, and lymphocyte proliferation and differentiation. Our research provides a theoretical basis for the use of *R. mucilaginosa* ZTHY2 as an immunopotentiator and a means of attenuating CTX-induced immunosuppression, and it gives an application perspective for exploration in other fields.

## 2. Materials and Methods

### 2.1. Microbial Strain, Antibodies, and Reagents

*R. mucilaginosa* ZTHY2 was isolated and identified by our team and was stored in the China Center for Type Culture Collection, CCTCC (Wuhan, China). The preservation number is M2015296. *Lactobacillus acidophilus* (BNCC185342) was purchased from BeNa Culture Collection (Beijing, China). Antibodies of FITC anti-mouse CD4 (100509), PE anti-rat CD3 (100205), APC anti-mouse CD8a (100711), PerCP anti-mouse CD19 (115531), APC anti-mouse CD20 (152107), and Zombie NIR™ Fixable Viability Kit were all purchased from Biolegend (San Diego, CA, USA). CTX was purchased from Yuanye biotechnology Co., Ltd. (Shanghai, China). ConA and LPS were purchased from Solarbio Life Sciences (Beijing, China).

### 2.2. Animals, Experimental Design, and Management

A total of 72 SPF male mice (C57BL/6), five weeks old, were used in a 28-day feeding trial (six to nine weeks old during the study). The mice, with an initial average body weight (BW) of 18.25 ± 2 g, were sourced from Guangzhou Dean Gene biological Co., Ltd. (Guangzhou, China). Mice were randomly and equally divided into six groups, namely the normal control group (N), immunosuppressive model group (IM), and positive control group (PC), and the *R. mucilaginosa* ZTHY2 treatment groups with low, medium, and high concentration, respectively (Rl, Rm, Rh). Each group had two replicates with six mice in each replicate. From 1 to 21 days, the N and IM groups were treated with 0.5 mL saline via gavage at each day. The PC group was treated with LA at 2 × 10^9^ CFU/mL, 0.5 mL. The Rl, Rm, and Rh groups were treated with 0.5 mL of *R. mucilaginosa* ZTHY2 with concentrations of 2 × 10^7^ CFU/mL, 2 × 10^8^ CFU/mL, and 2 × 10^9^ CFU/mL, respectively. From the 22nd–28th day, except for the N group, CTX was treated via gavage at 50 mg/kg for the other groups (Figure 1A). All the mice were kept in an environmentally controlled room, with the temperature maintained at (25 ± 1) °C, and the relative humidity was 65–85%. The mice had free access to water and feed. The formal experiment was carried out after one week of adaptation. All experimental protocols were approved by the Animal Ethics Committee of Guangdong Ocean University, (IACUC No. GDOU-LAE-2020-007), and the experiment was performed according to the ethical guidelines of the European Community guidelines.

### 2.3. Body Weight and Immunity Organ Indices

The mice were weighed on day 0 (before the experiment), day 22 (before treatment with CTX) and day 29 (before dissecting the mice). At the 29th day, all the mouse blood was collected, followed by neck amputation, and studied with the gross anatomy to collect immune organs (the spleen, thymus, and mesenteric lymph nodes). The spleen and thymus were weighed to calculate the immune organ index as follows.
$$Immune\;organ\;index\;(mg/g)\;=\;\frac{organ\;weight\;(mg)}{body\;weight\;(g)}$$

### 2.4. Determination of Hematological Indices

Mouse blood was collected from the eyeballs, of which 50 μL anticoagulant blood from each mouse was analyzed using an automatic blood cell analyzer (URIT-5180, Medical Electronic Co., Ltd., Guilin, China) to analyze the changes in leukocytes (WBCs), monocytes (MONs), lymphocytes (LYMs), neutrophils (NEUs), basophils (BASOs), and eosinophils (EOSs).

### 2.5. Enzyme-Linked Immunosorbent Assay (ELISA)

Blood was collected without anticoagulant, and the serum samples were obtained by centrifugation at 3000 rpm (10 min, 4 °C) after coagulation. The immunoglobulins IgA, IgG, and IgM; the interleukins IL-2, IL-4, and IL-6; tumor necrosis factor-α (TNF-α); interferon-γ (IFN-γ); and clusters of differentiation CD4, CD8, CD19, and CD20 in the serum were analyzed using commercial ELISA kits for mice (catalog numbers: IgA, QS42791; IgG, QS49338; IgM, QS42794; IL-2, QS42903; IL4, QS42901; IL-6, QS42899; TNF-α, QS42868; IFN-γ, QS42918; CD4, QS40227; CD8, QS440266; CD19, QS445871; CD20, QS40582; Beijing Gersion Bio-Technology Co., Ltd., Beijing, China).

### 2.6. Antioxidant Capacity Analysis

To analyze the antioxidant capacity of mice, the serum samples were obtained by centrifugation at 3000 rpm (10 min, 4 °C) after blood coagulation. Afterward, MDA contents were analyzed using commercial kits (catalog number: S0131S; Beyotime, Shanghai, China). Antioxidant enzyme activity, including SOD and CAT, was analyzed using commercial kits (catalog numbers: SOD, BC0170; CAT, BC0200; Solarbio Life Sciences, Beijing, China). Glutathione peroxidase (GSH-Px) was analyzed using commercial kits (catalog number: A005-1-1. Nanjing Jiancheng Bioengineering Institute, Nanjing, China).

### 2.7. Single-Cell Suspension of Spleen, Thymus, and Mesenteric Lymph Nodes

The spleen, thymus, and mesenteric lymph nodes were collected and digested with pancreatic enzyme until the tissue became mushy. Then, digestion was terminated with twice the volume of cell culture medium. Next, the single-cell suspension was obtained with a 70 μm cell filter.

### 2.8. Intracellular Reactive Oxygen Species (ROS) of Spleen

The spleen single-cell suspension was centrifuged at 1200 rpm for 5 min and re-suspended with 1:500 diluted 2,7-dichlorodihydrofluorescein diacetate (DCFH-DA) for three hours, then washed by PBS and detected using a FACSVantage cytometer (SE, Becton Dickinson, Franklin Lakes, NJ, USA), with the excitation setting at 488 nm, and signals were acquired at the FL-2 channel. At least 10,000 cells per sample were acquired in histograms, and the data were analyzed with CytExpert (2.3.0.84) software. The DCFH-DA was purchased from Sigma (St. Louis, MO, USA) and dissolved in ethanol to produce a 1 mmol/L stock solution.

### 2.9. Proliferation of T and B Lymphocytes in Spleen and Thymus In Vitro

The spleen and thymus single-cell suspension were centrifuged at 500× *g* for 5 min, then suspended with red blood cell lysate for 3 min, and then centrifuged, and 2 mL of RPMI-1640 culture medium was used for suspension. The cell was inoculated in 96-well cell culture plates, with 1.0 × 10^4^ cell/well. ConA was added to the spleen single-cell suspension to stimulate T cell proliferation, and LPS was added to the thymus single-cell suspension to stimulate B cell proliferation. The cell proliferation was detected using CCK-8 (Abmole, Houston, TX, USA) following the manufacturer’s instructions.

### 2.10. Flow Cytometry Detection of T and B Cell Species of Spleen, Thymus, and Mesenteric Lymph Nodes

A single-cell suspension of 100 μL was taken and adjusted to a total amount of 1 × 10^6^ cells, centrifuged at 500× *g* for 5 min, re-suspended with Zombie NIR dye, and incubated at room temperature for 15–30 min away from light. PBS-BSA was used to clean the cells, and a 100 μL suspension was retained after centrifugation. Next, 1 μL FITC-CD4 antibody, 1.25 μL ACP-CD8 antibody, and 1.25 μL PE-CD3 antibody were added to the cells and incubated at room temperature for 15–30 min to detect T cell types, while 1 μL FITC-CD19 antibody and 1.25 μL CD20 antibody were added to the cells and incubated at room temperature for 15–30 min to detect B cell types. Moreover, 300 μL red cell lysate was added to each tube incubated away from light for 10 min, centrifuged at 300× *g* for 5 min, and then 1 mL PBS-BSA was added to the mixture, which was centrifuged again to discard the supernatant, followed by machine detection. The cells were detected using a flow cytometer.

### 2.11. Statistical Analysis

The statistical analysis of the data was carried out using SPSS 22.0. Statistical significance among multiple comparisons was determined via one-way ANOVA, followed by Ducan’s test using GraphPad Prism 5. The experimental results are expressed as mean ± standard deviation; a *p*-value less than 0.05 was considered to indicate a significant difference, and a *p*-value less than 0.01 was considered to indicate an extremely significant difference.

## 3. Results

### 3.1. Effects of R. mucilaginosa ZTHY2 on Body Weight and Immune Organ Index in CTX-Induced Immunosuppressed Mice

To address whether *R. mucilaginosa* ZTHY2 was able to enhance or protect immunity from suppression, we treated CTX-induced immunosuppressed mice and determined animal body weight, followed by isolation of the spleen and thymus for studies, as depicted in Figure 1A. As shown in Table 1, there was no significant difference in body weight among all groups at the initial weight (Day 0) and after gavage with *R. mucilaginosa* ZTHY2 or LA for 21 days (Day 22). After treatment with CTX for one week, the body weight of the IM group was significantly decreased (*p* < 0.01). Compared with the IM group, the body weight of the R groups and PC group was significantly increased (*p* < 0.05) (Day 29). As shown in Figure 1B,C, compared with the N group, the spleen and thymus indices in the IM group were significantly decreased (*p* < 0.01). Compared with the IM group, the spleen and thymus indices in the PC and R groups were significantly increased (*p* < 0.01). There was no significant difference between the Rh and N groups (*p* > 0.05), indicating that *R. mucilaginosa* ZTHY2 could effectively attenuate the atrophy of the spleen and thymus induced by CTX.

### 3.2. Effect of R. mucilaginosa ZTHY2 on Hematological Indices in CTX-Induced Immunosuppressed Mice

The results of biochemical analysis of the mice’s blood showed that, as shown in Table 2, compared with the N group, the numbers of WBCs, LYMs, MONs, NEUs, and EOSs were significantly decreased in the IM group (*p* < 0.01). Compared with the IM group, the numbers of WBCs, LYMs, MONs, and NEUs in the R groups and the PC group were significantly increased (*p* < 0.05). There was no difference in any of the detected indicators between the Rh group and the PC group. This showed that *R. mucilaginosa* ZTHY2 can attenuate the blood toxicity caused by CTX.

### 3.3. R. mucilaginosa ZTHY2 Increased the Level of Immunoglobulin in CTX-Induced Immunosuppressed Mice

To detect the effect of *R. mucilaginosa* ZTHY2 on immunoglobulin levels in CTX-induced immunosuppressed mice, the ELISA method was adopted to analyze the serum immunoglobulins IgA, IgG, and IgM in mice. As shown in Figure 2, compared with the N group, the levels of IgA (Figure 2A), IgG (Figure 2B), and IgM (Figure 2C) in serum in the IM group were significantly decreased (*p* < 0.01). Compared with the IM group, the levels of IgG, IgA, and IgM in the Rl group were significantly increased (*p* < 0.05) and extremely significantly increased (*p* < 0.01) in the Rm, Rh, and PC groups. The promotion effect of immunoglobulin by *R. mucilaginosa* ZTHY2 was dose-dependent. This showed that *R. mucilaginosa* ZTHY2 can attenuate the adverse effect of CTX on the immunoglobulins.

### 3.4. R. mucilaginosa ZTHY2 Increased the Secretion of Immune Cytokines in CTX-Induced Immunosuppressed Mice

To detect the effect of *R. mucilaginosa* ZTHY2 on immune cytokines in immunosuppressed mice, the ELISA was used to analyze the IL-2, IL-4, IL-6, TNF-α, and IFN-γ in the serum. As shown in Figure 3, the protein levels of IL-2 (Figure 3A), IL-4 (Figure 3B), IL-6 (Figure 3C), TNF-α (Figure 3D), and IFN-γ (Figure 3E) in the IM group were significantly decreased compared with the N group (*p* < 0.01). Compared with the IM group, protein levels of L-2, IL-4, IL-6, TNF-α, and IFN-γ were significantly increased in the Rl group (*p* < 0.05) and extremely significantly increased in the Rm, Rh, and PC groups (*p* < 0.01), which indicated that *R. mucilaginosa* ZTHY2 could promote the secretion of immune cytokines in serum and attenuated the immunosuppressant response induced by CTX.

### 3.5. R. mucilaginosa ZTHY2 Increased the Levels of CDs in CTX-Induced Immunosuppressed Mice

In order to investigate the effect of *R. mucilaginosa* ZTHY2 on leukocyte differentiation and maturation, the ELISA was used to analyze CD4, CD8, CD19, and CD20 in the serum. As shown in Figure 4, the protein levels of CD4 (Figure 4A), CD8 (Figure 4B), CD19 (Figure 4C), and CD20 (Figure 4D) in the IM group were significantly decreased compared with the N group (*p* < 0.01). The protein levels of CD4, CD8, CD19, and CD20 in the Rl group were significantly increased compared with the IM group (*p* < 0.05). The protein levels in the Rm, Rh, and PC groups were significantly increased (*p* < 0.01). The protein levels of CD4 and CD20 in the Rh group were not significantly different from those in the N group (*p* > 0.05). These results indicated that *R. mucilaginosa* ZTHY2 could increase the proportion of CD4, CD8, CD19, and CD20 in immunosuppressed mice, thereby regulating the immune function of the body.

### 3.6. R. mucilaginosa ZTHY2 Promoted the Proliferation of T and B Lymphocytes of Immunosuppressed Mice Induced by CTX In Vitro

To determine whether *R. mucilaginosa* ZTHY2 could promote lymphocyte proliferation, ConA, known to activate the T cells [43], and LPS, widely used to activate the B cells [44], were used to induce the proliferation of lymphocytes in vitro. As shown in Figure 5, the proliferation of spleen T (Figure 5A) and B (Figure 5B) lymphocytes and thymus T (Figure 5C) and B (Figure 5D) lymphocytes in the MI groups was significantly decreased compared with the N group (*p* < 0.01). Compared with the IM group, the proliferation was significantly increased in the Rl (*p* < 0.05), Rm and Rh (*p* < 0.01), and PC groups (*p* < 0.01), and the trend was dependent on the concentration. This showed that the proliferation of T and B lymphocytes in the spleen and thymus of immunosuppressed mice was enhanced by *R. mucilaginosa* ZTHY2.

### 3.7. Effects of R. mucilaginosa ZTHY2 on T Cell and B Cell Subsets in CTX-Induced Immunosuppressed Mice

To further analyze the effects of *R. mucilaginosa* ZTHY2 on T cell and B cell subsets in CTX-induced immunosuppressed mice, the levels of CD3, CD4, and CD8 were detected via flow cytometry to verify the T cell differentiation, and CD19 and CD20 were detected to verify the B cell differentiation.

#### 3.7.1. Effects of *R. mucilaginosa* ZTHY2 on T Cell and B Cell Subsets of Spleen in CTX-Induced Immunosuppressed Mice

The analysis of lymphocyte subsets in the spleen showed that compared with the N group, the percentages of CD3^+^, CD4^+^, CD8^+^, CD19^+^, and CD20^+^ lymphocytes and the ratio of CD4^+^/CD8^+^ in the IM group were significantly decreased (*p* < 0.01). Compared with the IM group, the percentages of CD3^+^, CD4^+^, CD8^+^, CD19^+^, and CD20^+^ lymphocytes and the ratio of CD4^+^/CD8^+^ were significantly increased in the R groups and the PC group (*p* < 0.05). There was no significant difference in the percentages of CD4^+^, CD8^+^, and CD19^+^ between the R groups and PC group (*p* > 0.05) and CD3^+^ and CD20^+^ cells between the Rm and Rh groups compared to the PC group (*p* > 0.05). There was no significant difference in CD3^+^, CD4^+^, and CD20^+^ cells between the Rh and N groups (*p* > 0.05) (Figure 6 and Table 3). The results indicated that *R. mucilaginosa* ZTHY2 could significantly stimulate both T and B lymphocyte differentiation of spleen in CTX-induced immunosuppressed mice, and gavage with *R. mucilaginosa* ZTHY2 at the concentration of 2 × 10^8^ CFU/mL had the same effect as LA at 2 × 10^9^ CFU/mL.

#### 3.7.2. Effects of *R. mucilaginosa* ZTHY2 on T Cell and B Cell Subsets of Thymus in CTX-Induced Immunosuppressed Mice

The analysis of lymphocyte subsets in the thymus showed that compared with those in the N group, the percentage of CD3^+^, CD4^+^, CD8^+^, CD19^+^, and CD20^+^ cells and the ratio of CD4^+^/CD8^+^ in the IM group were significantly decreased (*p* < 0.01). Compared with those in the IM group, the percentages of CD3^+^, CD4^+^, CD8^+^, CD19^+^, and CD20^+^ lymphocytes in the R groups and PC group were significantly increased (*p* < 0.05). There was no significant difference in CD3^+^, CD4^+^, CD8^+^, CD19^+^, CD20^+^ percentage and the ratio of CD4^+^/CD8^+^ among the Rm, Rh, and PC groups (*p* > 0.05) and no significant difference in CD3^+^, CD4^+^, and CD19^+^ percentage between Rh and N (*p* > 0.05) (Figure 7 and Table 4). The results indicated that *R. mucilaginosa* ZTHY2 could significantly stimulate lymphocyte differentiation of the thymus in CTX-induced immunosuppressed mice, and gavage with *R. mucilaginosa* ZTHY2 at the concentration of 2 × 10^8^ CFU/mL had the same effect as LA at 2 × 10^9^ CFU/mL.

#### 3.7.3. Effects of *R. mucilaginosa* ZTHY2 on T Cell and B Cell Subsets of Mesenteric Lymph Nodes in CTX-Induced Immunosuppressed Mice

The analysis of lymphocyte subsets in mesenteric lymph nodes showed that compared with those in the N group, the percentage of CD3^+^, CD4^+^, CD8^+^, CD19^+^, and CD20^+^ lymphocytes and the ratio of CD4^+^/CD8^+^ in the IM group were significantly decreased (*p* < 0.01). Compared with those in the IM group, the percentages of CD3^+^, CD4^+^, CD8^+^, and CD20^+^ lymphocytes in the R groups and PC group were significantly increased (*p* < 0.01). There was no significant difference in CD3^+^, CD4^+^, CD8^+^, and CD20^+^ lymphocytes percentage among the R groups and PC group (*p* > 0.05). There was no significant difference in CD3^+^, CD4^+^, and the ratio of CD4^+^/CD8^+^ among the Rm, Rh, and N groups (*p* > 0.05) (Figure 8 and Table 5). The results indicated that *R. mucilaginosa* ZTHY2 could significantly stimulate lymphocyte differentiation of mesenteric lymph nodes in CTX-induced immunosuppressed mice, and gavage with *R. mucilaginosa* ZTHY2 at the concentration of 2 × 10^8^ CFU/mL had the same effect as LA at 2 × 10^9^ CFU/mL.

### 3.8. Effects of R. mucilaginosa ZTHY2 on Antioxidant Capacity of Spleen in CTX-Induced Immunosuppressed Mice

The antioxidant capacity of the body is also closely related to immune function and health, so in order to investigate the effect of *R. mucilaginosa* ZTHY2 on the oxidation capacity, the MDA, ROS, CAT, GSH, and SOD of the spleen were detected. As shown in Figure 9, compared with those in the N group, the levels of ROS (Figure 9A) and MDA (Figure 9E) in the IM group were significantly increased (*p* < 0.01), while the levels of CAT (Figure 9B), SOD (Figure 9C), and GSH (Figure 9D) were significantly decreased (*p* < 0.01). The results were significantly reversed in both the R groups and the PC group and dependent on the concentration of *R. mucilaginosa* ZTHY2 (*p* < 0.01). There were no significant difference in SOD, CAT, and GSH contents between the Rh group and the N group (*p* > 0.05). The results indicated that *R. mucilaginosa* ZTHY2 could effectively increase the content of GSH, SOD, and CAT in immunosuppressed mice, remove excess ROS in the spleen (*p* < 0.01), and then reduce MDA produced by peroxidation.

## 4. Discussion

Immunosuppression is a condition characterized by temporary or permanent immune dysfunction. In intensive livestock and poultry farms, immunosuppression is easy to see, and feed additives that can enhance immunity and attenuate immunosuppression are needed. *R. mucilaginosa* ZTHY2 exhibits immunostimulatory properties and regulates the gastrointestinal microbiome [42]. Furthermore, our study confirmed its ability to enhance immune response and antioxidant capacity in mice under immunosuppressive conditions. *R. mucilaginosa* ZTHY2 reduced weight loss and the atrophy of immune organs, decreased proliferation and differentiation of immune cells, and reduced production of immune factors induced by CTX. Our results show that *R. mucilaginosa* ZTHY2 has good potential for use as an immunomodulator. At present, there are many studies exploring immunomodulators, such as peptides from soybean, nucleic acid combined with microRNA, extracts of plants and medicinal herbs, and peptides from aquatic organisms [45,46,47,48,49]. However, *R. mucilaginosa* ZTHY2 has a great advantage in that it is widely available in the ocean, it is easy to isolate and culture, and the culture cost is low; hence, it has good application prospects and is worth further research.

The activity of LA in stimulating the immune system has been verified in a variety of species. The immunomodulatory effects of LA were demonstrated in mice as early as 1986 [50]. Subsequently, they have been confirmed in a variety of animals such as laying hens, broiler chickens, zebrafish, and weaned calves [51,52,53,54]. Different strains of LA ameliorated the CTX-induced immunosuppression in mice, and the protective function of LA from CTX was reflected in LA reversing the body and spleen weights, blood RBC and WBC levels, and splenocyte and bone-marrow cells; increased T cell proliferation; induced expression of IL-2 and IFN-gamma in T cells; the restored phagocytosis of macrophages in mice; and cytotoxicity of NK and cytotoxic T cells [55,56,57,58]. Accordingly, LA was used as a positive control in our experiment.

The spleen and thymus are the important immune organs, and the spleen and thymus indices can reflect the immune function of the body. Splenic lymphocyte proliferation is an important biomarker of activation of cellular and humoral immune responses. T and B lymphocyte proliferation is usually induced by mitogen ConA and LPS [59,60]. Studies have shown that CTX decreased the immune organ indices and inhibited the T and B lymphocyte proliferation [61,62]. In this study, the spleen and thymus indices decreased significantly, and the proliferation ability of T and B lymphocytes decreased; these indicated that the immunosuppressive model was successfully established. With the protection of *R. mucilaginosa* ZTHY2, the spleen and thymus indices of mice were improved in a concentration-dependent manner, suggesting that *R. mucilaginosa* ZTHY2 could protect against the damage of CTX to the spleen and thymus.

The blood actively engages in metabolic reactions and plays a crucial role in maintaining homeostasis within the body. White blood cells are composed of lymphocytes, neutrophils, monocytes, eosinophils, and basophils. Studies have shown that CTX induces a significant decrease in the number of immune cells, such as leukocytes, monocytes, and lymphocytes, resulting in low immunity [63]. In this experiment, the blood indices of mice in the IM group were significantly decreased compared to those in the N group, indicating that CTX inhibited the hemocytogenesis in mice. On the contrary, the blood indices of mice in the R groups were significantly improved, especially the leukocytes, which are an important part of the body’s defense system, indicating that *R. mucilaginosa* ZTHY2 can enhance the body’s immune function and remove the toxic effect of CTX on blood cells.

The innate and adaptive immune systems are further divided into cellular immunity and humoral immunity. T and B lymphocytes are responsible for cellular immunity and humoral immunity, respectively [64,65]. Classically, based on the surface markers of T cells, T lymphocyte subsets can be divided into CD3^+^, CD4^+^, and CD8^+^ T cells. CD3^+^ T lymphocytes exist on the surface of all T lymphocytes and closely bind to T cell receptors to form a TCR–CD3 complex containing eight peptide chains, which jointly participate in the recognition and signaling of antigens by T cells [66,67]. CD4^+^ represents the cellular immune secretion function and T helper cell (Th cell) subsets. Th cells induce cytokine differentiation and mediate cellular and humoral immune responses. CD8^+^ T cells can kill labeled cells and inhibit antigen-presenting cells, and the percentage of CD8^+^ is a measure of the killing ability of cellular immune and tissue damage [68,69]. Generally, the number of T cell subsets is relatively stable. When the number of T cells or the ratio of CD4^+^/CD8^+^ is reduced, it often indicates immunosuppression, like malignant tumors or viral infections, and when the ratio is increased, autoimmune diseases may occur [70]. In addition, the immune function of the body can be evaluated by analyzing the content of CD4 and CD8 molecules. CD4 can be further divided into Th1 and Th2 subsets; Th1 cells produce cytokines such as IL-2, TNF-α, and IFN-γ, and Th2 cells produce IL-4, IL-5, IL-6, IL-10, and IL-13 [71]. IL-2 can regulate the function of B lymphocytes and natural killer cells, which reflects the function of cellular immunity [72]. IFN-γ and IL-2 have a synergistic effect on inhibiting viral replication, activating macrophages, and enhancement the cellular immune function [73]. TNF-α can selectively kill tumor cells, stimulate lymphocyte proliferation, and activate NK cells [74]. IL-4 can promote IgE-mediated humoral immune response by activating B cells to synthesize IgE and transform IgG into IgE [75]. IL-6 can induce B cell differentiation and maturation and participate in immune response and inflammation [76]. Under physiological conditions, Th1/Th2 cells in the body are in a state of dynamic balance. When the body is dysfunctional, the levels of Th1 and Th2 cells are adjusted to ensure the normal function. Studies have shown that when animals suffer from CTX, the secretion levels of CD3^+^, CD4^+^, CD8^+^, and CD4^+^/CD8^+^ are reduced, resulting in an imbalance of Th1/Th2 cell proportion, decreased immune function, and induced secondary infection [77]. The findings of our study are consistent with this conclusion. The results demonstrated a significant decrease in the levels of immune cytokines in the serum of mice in the IM group, as well as the secretion of CD3^+^, CD4^+^, CD8^+^ lymphocytes, and the ratio of CD4^+^/CD8^+^ in spleen, thymus, and mesenteric lymph nodes. Gavage with *R. mucilaginosa* ZTHY2 could significantly increase the levels of immune cytokines, CD4, and CD8 molecules in immunosuppressed mice, increased the percentage of CD3^+^, CD4^+^, CD8^+^ lymphocytes and the ratio of CD4^+^/CD8^+^, restoring the balance of Th1/Th2 cells and thus enhancing the cellular immune function of immunosuppressed mice.

CD19 is an antigen involved in B cell proliferation, differentiation, and antibody production and can promote the signal transduction of B cell receptors [78]. CD20 is a surface marker of B cells and plays an important regulatory role in the differentiation and proliferation of B cells [79]. B lymphocytes produce immunoglobulin, an important component of humoral immunity, in response to antigen stimulation. IgA has an anti-infection effect, IgG is the main immunoglobulin produced by B lymphocyte activation, and IgM is the earliest immunoglobulin produced by the immune system in response to antigens, and its proportion in the blood is only 10% [80,81]. In this study, gavage of *R. mucilaginosa* ZTHY2 in immunosuppressed mice increased the level of IgA, IgG, and CD19 and CD20 content in serum; promoted the secretion of CD19^+^ and CD20^+^ B lymphocytes; and regulated humoral immune function. Our results were consistent with those of other species like Penaeus vannamei and juvenile Atlantic salmon [82,83].

External factors or a decline in autoimmunity may cause immune suppression, leading to a decrease in immune and antioxidant function [84]. When the antioxidant capacity is reduced, excess ROS are generated, and their reaction with unsaturated fatty acids produces MDA, a lipid peroxidation product. The content of MDA is an important index to evaluate oxidative damage [85]. SOD is involved in regulating the redox reaction and maintaining the oxidation function by scavenging oxygen free radicals [86]. CAT is mainly present in chloroplasts of plants and in the liver and red blood cells of animals, in especially high concentrations in the liver, and it can decompose hydrogen peroxide into O_2_ and H_2_O. If hydrogen peroxide cannot be decomposed in time, it causes serious damage to the body [87,88]. There are two forms of glutathione: reduced glutathione and oxidized glutathione. Reduced glutathione, which is generally 90–95% in content, works in concert with other cellular antioxidants to remove free radicals; it can thereby relieve oxidative stress [89]. Studies on *R. mucilaginosa* in Nile tilapia showed that SOD activity was improved both in serum and the liver, as well as the T-AOC in the serum, and the MDA content was notably reduced in the liver [10]. Our results showed that *R. mucilaginosa* ZTHY2 could decrease ROS and MDA levels in the spleen and increase SOD and CAT activities in immunosuppressed mice, which is consistent with the results in tilapia. These results suggest that *R. mucilaginosa* ZTHY2 can promote antioxidant levels in both normal and immunosuppressed animals.

## 5. Conclusions

Mice were fed with *R. mucilaginosa* ZTHY2 for 28 days and then treated with CTX. The results showed that *R. mucilaginosa* ZTHY2 could enhance immune function and antioxidant capacity against CTX. Our study provides theoretical support for the use of *R. mucilaginosa* ZTHY2 as a feed additive in livestock and poultry as an immunomodulator to improve disease resistance and adaptability, as well as a reagent to attenuate the adverse effects of CTX.

## Figures and Tables

**Figure 1 animals-13-03376-f001:**
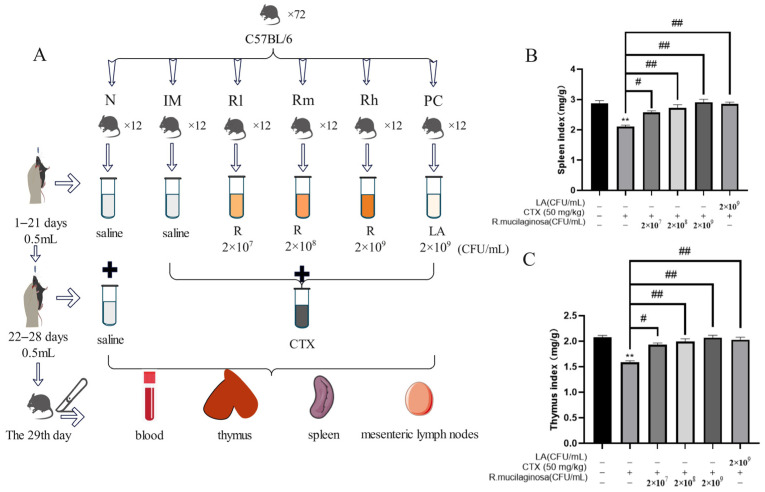
Effects of *R. mucilaginosa* ZTHY2 on the spleen and thymus indices in CTX-induced immunosuppressed mice. (**A**) The protocol of animal experiments. (**B**) The spleen index. (**C**) The thymus index. N: normal control group. IM: immunosuppressive model. R: *R. mucilaginosa* ZTHY2. Rl: *R. mucilaginosa* ZTHY2 with low concentration. Rm: *R. mucilaginosa* ZTHY2 with medium concentration. Rh: *R. mucilaginosa* ZTHY2 with high concentration; PC: positive control. CTX: cyclophosphamide. LA: *Lactobacillus acidophilus.* Comparison with N group, differences are indicated by “*”, ** (*p* < 0.01). Comparison with IM group, differences are indicated by “#”, # (*p* < 0.05), ## (*p* < 0.01).

**Figure 2 animals-13-03376-f002:**
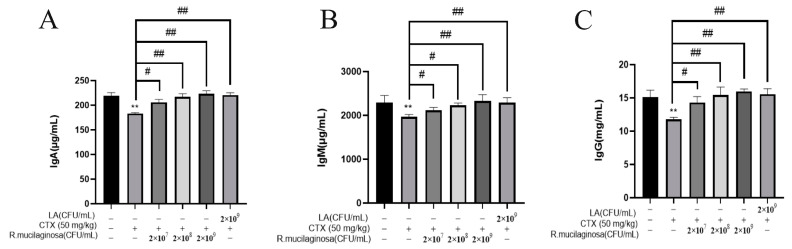
*R. mucilaginosa* ZTHY2 increased the level of immunoglobulin in CTX-induced immunosuppressed mice. The IgG, IgA, and IgM in the serum were detected by the ELISA. (**A**) The level of IgA. (**B**) The level of IgM. (**C**) The level of IgG. Comparison with N group, differences are indicated by “*”, ** (*p* < 0.01); comparison with IM group, differences are indicated by “#”, # (*p* < 0.05), ## (*p* < 0.01).

**Figure 3 animals-13-03376-f003:**
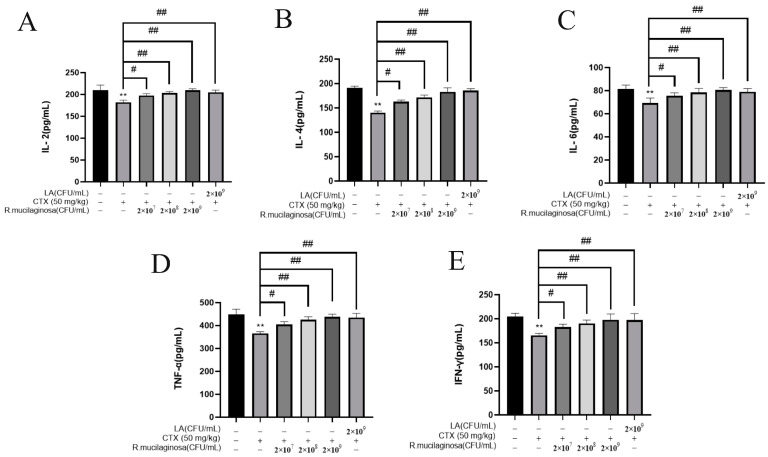
*R. mucilaginosa* ZTHY2 increased the secretion of immune cytokines in CTX-induced immunosuppressed mice. The IL-2, IL-4, IL-6, TNF-α, and IFN-γ in the serum were detected by the ELISA. (**A**) The level of IL-2. (**B**) The level of IL-4. (**C**) The level of IL-6. (**D**) The level of TNF-α. (**E**) The level of IFN-γ. Comparison with N group, differences are indicated by “*”, ** (*p* < 0.01); comparison with IM group, differences are indicated by “#”, # (*p* < 0.05), ## (*p* < 0.01).

**Figure 4 animals-13-03376-f004:**
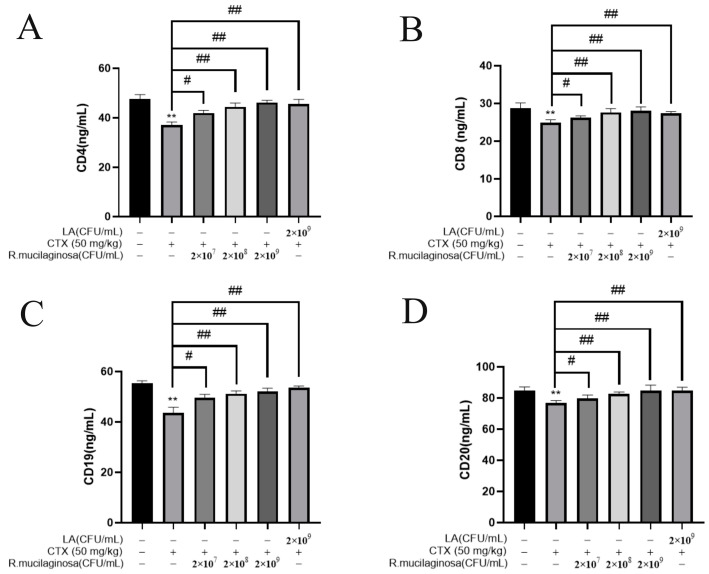
*R. mucilaginosa* ZTHY2 increased the levels of CDs in CTX-induced immunosuppressed mice. The CD4, CD8, CD19, and CD20 in the serum were detected by the ELISA. (**A**) The level of CD4. (**B**) The level of CD8. (**C**) The level of CD19. (**D**) The level of CD20. Comparison with N group, differences are indicated by “*”, ** (*p* < 0.01); comparison with IM group, differences are indicated by “#”, # (*p* < 0.05), ## (*p* < 0.01).

**Figure 5 animals-13-03376-f005:**
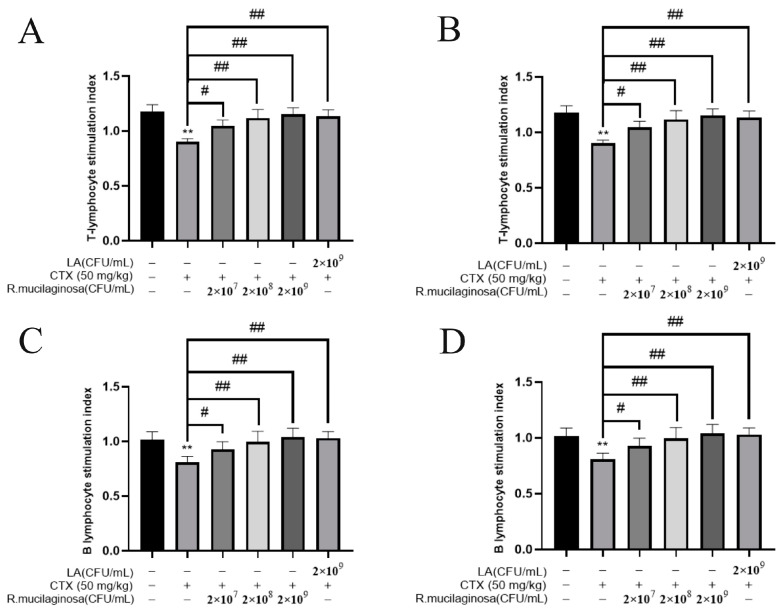
*R. mucilaginosa* ZTHY2 promoted the proliferation of T and B lymphocytes of immunosuppressed mice induced by CTX in vitro. Single-cell suspensions of the spleen and thymus were prepared, and lymphocyte proliferation was detected using CCK-8 after stimulation with ConA and LPS. (**A**) Proliferation of T lymphocytes of the spleen stimulated by ConA. (**B**) Proliferations of B lymphocytes of the spleen stimulated by ConA. (**C**) Proliferations of T lymphocytes of the thymus stimulated by LPS. (**D**) Proliferations of B lymphocytes of the thymus stimulated by LPS. Comparison with N group, differences are indicated by “*”, ** (*p* < 0.01); comparison with IM group, differences are indicated by “#”, # (*p* < 0.05), ## (*p* < 0.01).

**Figure 6 animals-13-03376-f006:**
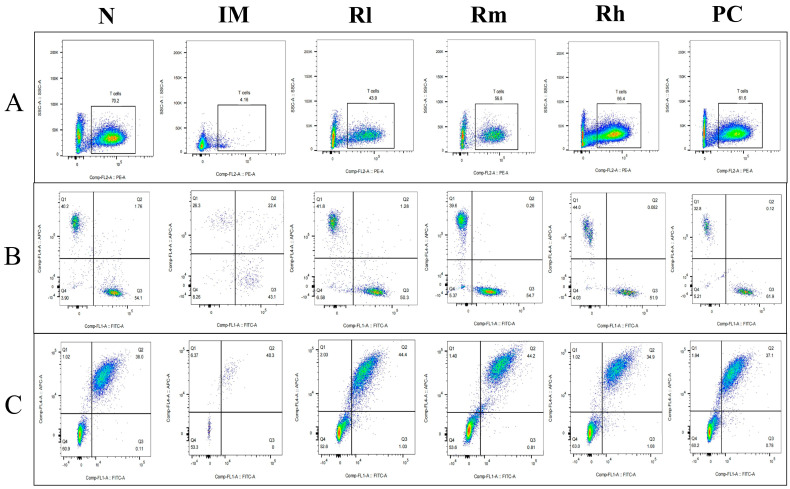
Expression of marker molecules of lymphocyte subpopulations in mouse spleen. Single-cell suspensions of spleen were prepared, and CD3^+^, CD4^+^, CD8^+^, CD19^+^, and CD20^+^ lymphocytes were detected using a flow cytometer. (**A**) Detection of CD3^+^ lymphocytes. (**B**) Detection of CD4^+^ and CD8^+^ lymphocytes. (**C**) Detection of CD19^+^ and CD20^+^ lymphocytes. N: normal control group. IM: immunosuppressive model group. Rl: *R. mucilaginosa* ZTHY2 with low concentration. Rm: *R. mucilaginosa* ZTHY2 with medium concentration. Rh: *R. mucilaginosa* ZTHY2 with high concentration. PC: positive control.

**Figure 7 animals-13-03376-f007:**
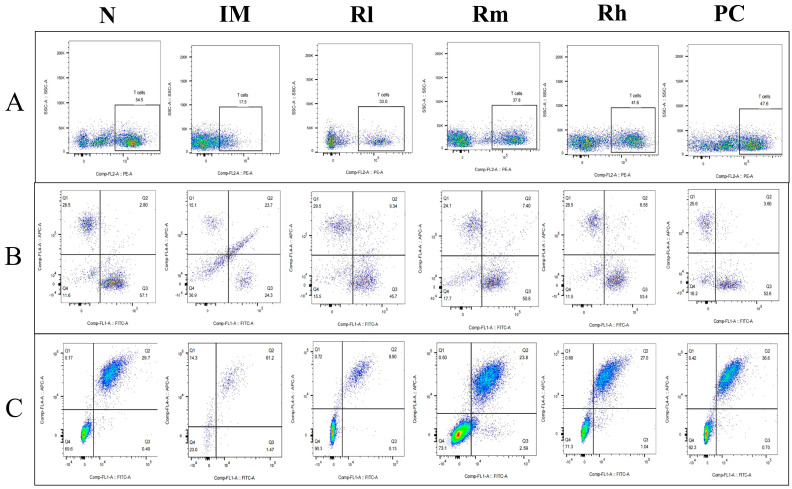
Expression of marker molecules of lymphocyte subpopulations in mice thymus. Single-cell suspensions of thymus were prepared, and CD3^+^, CD4^+^, CD8^+^, CD19^+^, and CD20^+^ lymphocytes were detected using a flow cytometer. (**A**) Detection of CD3^+^ lymphocytes. (**B**) Detection of CD4^+^ and CD8^+^ lymphocytes. (**C**) Detection of CD19^+^ and CD20^+^ lymphocytes. N: normal control group. IM: immunosuppressive model group. Rl: *R. mucilaginosa* ZTHY2 with low concentration. Rm: *R. mucilaginosa* ZTHY2 with medium concentration. Rh: *R. mucilaginosa* ZTHY2 with high concentration. PC: positive control.

**Figure 8 animals-13-03376-f008:**
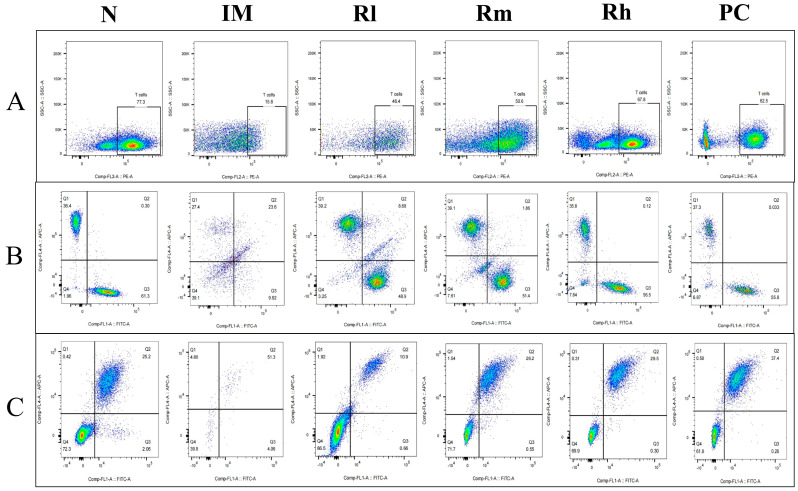
Expression of marker molecules of lymphocyte subpopulations in mice mesenteric lymph nodes. Single-cell suspensions of mesenteric lymph nodes were prepared, and CD3^+^, CD4^+^, CD8^+^, CD19^+^, and CD20^+^ lymphocytes were detected using a flow cytometer. (**A**) Detection of CD3^+^ lymphocytes. (**B**) Detection of CD4^+^ and CD8^+^ lymphocytes. (**C**) Detection of CD19^+^ and CD20^+^ lymphocytes. N: normal control group. IM: immunosuppressive model group. Rl: *R. mucilaginosa* ZTHY2 with low concentration. Rm: *R. mucilaginosa* ZTHY2 with medium concentration. Rh: *R. mucilaginosa* ZTHY2 with high concentration. PC: positive control.

**Figure 9 animals-13-03376-f009:**
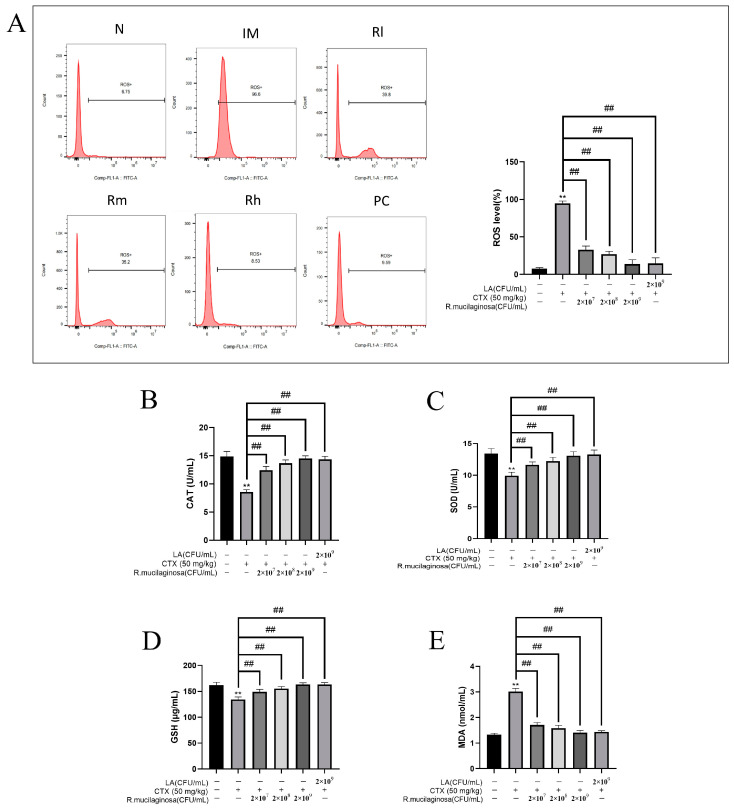
Effects of *R. mucilaginosa* ZTHY2 on antioxidant capacity in CTX-induced immunosuppressed mice. (**A**) ROS level of spleen in mice. (**B**) CAT level in the serum. (**C**) SOD level in the serum. (**D**) GSH level in the serum. (**E**) MDA level in the serum. N: normal control group. IM: immunosuppressive model. R: *R. mucilaginosa* ZTHY2. Rl: *R. mucilaginosa* ZTHY2 with low concentration. Rm: *R. mucilaginosa* ZTHY2 with medium concentration. Rh: *R. mucilaginosa* ZTHY2 with high concentration. PC: positive control. Comparison with the N group, differences are indicated by “*”, ** (*p* < 0.01); comparison with IM group, differences are indicated by “#”, ## (*p* < 0.01).

**Table 1 animals-13-03376-t001:** Changes in body weight of mice.

Group	Day 0 (g)	Day 22 (g)	Day 29 (g)
N	21.68 ± 0.74 ^a^	22.91 ± 0.48 ^a^	23.48 ± 0.47 ^Aa^
IM	21.00 ± 0.65 ^a^	22.83 ± 0.37 ^a^	21.66 ± 0.48 ^Bc^
Rl	20.58 ± 0.45 ^a^	22.73 ± 0.65 ^a^	22.70 ± 0.59 ^ABb^
Rm	21.03 ± 0.92 ^a^	22.75 ± 0.58 ^a^	22.90 ± 0.54 ^ABb^
Rh	21.33 ± 0.45 ^a^	22.90 ± 0.58 ^a^	23.68 ± 0.58 ^Aa^
PC	20.78 ± 0.59 ^a^	22.61 ± 0.81 ^a^	22.98 ± 0.81 ^ABb^

N: normal control group. IM: immunosuppressive model. Rl: *R. mucilaginosa* ZTHY2 with low concentration. Rm: *R. mucilaginosa* ZTHY2 with medium concentration. Rh: *R. mucilaginosa* ZTHY2 with high concentration. PC: positive control. Different lowercase letters in the same column indicate significant difference (*p* < 0.05). Different capital letters in the same column indicate extremely significant difference (*p* < 0.01). The same or no letters indicate non-significant difference (*p* > 0.05).

**Table 2 animals-13-03376-t002:** Hematological indices of mice.

Group	WBCs (10^9^/L)	LYMs (%)	MONs (%)	NEUs (%)	EOSs (%)
N	2.72 ± 0.93 ^Aa^	79.12 ± 1.81 ^Aa^	21.80 ± 9.10 ^Aa^	23.67 ± 1.78 ^ABa^	0.81 ± 0.14
IM	0.66 ± 0.03 ^Cc^	61.76 ± 0.84 ^Cc^	10.47 ± 4.85 ^Cc^	16.59 ± 10.48 ^Bb^	0.68 ± 0.10
Rl	1.21 ± 0.55 ^BCb^	68.91 ± 3.89 ^BCb^	16.64 ± 12.87 ^BCb^	20.25 ± 1.94 ^Aab^	0.72 ± 0.10
Rm	1.34 ± 0.40 ^BCb^	76.96 ± 1.72 ^ABb^	18.33 ± 6.87 ^Bb^	21.96 ± 7.52 ^Aab^	0.75 ± 0.04
Rh	1.81 ± 0.02 ^Bb^	78.74 ± 2.09 ^Aa^	20.25 ± 6.13 ^Aab^	23.24 ± 0.38 ^ABa^	0.78 ± 0.04
PC	1.87 ± 0.29 ^Bb^	77.24 ± 6.68 ^ABa^	21.21 ± 8.96 ^Aa^	22.41 ± 1.05 ^ABab^	0.81 ± 0.10

N: normal control group. IM: immunosuppressive model. Rl: *R. mucilaginosa* ZTHY2 with low concentration. Rm: *R. mucilaginosa* ZTHY2 with medium concentration. Rh: *R. mucilaginosa* ZTHY2 with high concentration. PC: positive control. WBCs: leukocytes; LYMs: lymphocytes; MONs: monocytes; NEUs: neutrophils; EOSs: eosinophilic granulocytes. Different lowercase letters in the same column indicate significant difference (*p* < 0.05). Different capital letters in the same column indicate extremely significant difference (*p* < 0.01). The same or no letters indicate non-significant difference (*p* > 0.05).

**Table 3 animals-13-03376-t003:** Detection of T and B lymph subpopulations in mouse spleen.

Group	T Lymph Subpopulations				B Lymph Subpopulations	
	CD3^+^ (%)	CD4^+^ (%)	CD8^+^ (%)	CD4^+^/CD8^+^	CD19^+^ (%)	CD20^+^ (%)
N	56.63 ± 7.75 ^Aa^	36.33 ± 2.03 ^Aa^	35.91 ± 6.37 ^Aa^	3.26 ± 2.73 ^Aa^	43.54 ± 4.01 ^Aa^	36.11 ± 3.97 ^Aa^
IM	8.24 ± 8.66 ^Cc^	8.17 ± 4.71 ^Bb^	6.71 ± 2.90 ^Cc^	0.98 ± 3.45 ^Cc^	26.82 ± 5.42 ^Cc^	21.55 ± 4.17 ^Bc^
Rl	49.35 ± 5.77 ^Bb^	27.5 ± 2.32 ^Aa^	25.98 ± 3.72 ^Bb^	1.07 ± 4.21 ^Cc^	33.02 ± 4.53 ^BCb^	25.80 ± 3.68 ^Bb^
Rm	52.02 ± 4.63 ^ABab^	34.45 ± 2.53 ^Aa^	27.42 ± 3.26 ^Bb^	1.64 ± 3.69 ^ABab^	35.03 ± 4.68 ^Bb^	27.90 ± 5.98 ^Ab^
Rh	55.28 ± 2.84 ^ABa^	35.35 ± 1.93 ^Aa^	31.33 ± 5.16 ^Bb^	2.30 ± 5.40 ^ABab^	37.56 ± 5.62 ^Bb^	31.57 ± 4.34 ^Aab^
PC	53.52 ± 5.35 ^ABab^	34.17 ± 3.04 ^Aa^	30.96 ± 4.72 ^Bb^	2.49 ± 4.87 ^ABab^	38.9 ± 5.61 ^Bb^	31.88 ± 3.94 ^Aab^

N: normal control group. IM: immunosuppressive model. Rl: *R. mucilaginosa* ZTHY2 with low concentration. Rm: *R. mucilaginosa* ZTHY2 with medium concentration. Rh: *R. mucilaginosa* ZTHY2 with high concentration. PC: positive control. Different lowercase letters in the same column indicate significant difference (*p* < 0.05). Different capital letters in the same column indicate extremely significant difference (*p* < 0.01). The same or no letters indicate non-significant difference (*p* > 0.05).

**Table 4 animals-13-03376-t004:** Detection of T and B lymph subpopulations in mouse thymus.

Group	T Lymph Subpopulations				B Lymph Subpopulations	
	CD3^+^ (%)	CD4^+^ (%)	CD8^+^ (%)	CD4^+^/CD8^+^	CD19^+^ (%)	CD20^+^ (%)
N	52.63 ± 2.75 ^Aa^	38.84 ± 12.03 ^Aa^	27.59 ± 2.41 ^Aa^	2.25 ± 4.813 ^Aa^	29.13 ± 1.64 ^Aa^	26.96 ± 4.42 ^Aa^
IM	10.24 ± 4.26 ^Cc^	11.81 ± 4.71 ^Cc^	9.01 ± 5.32 ^Cc^	0.88 ± 2.41 ^Cc^	12.82 ± 3.42 ^Cc^	10.99 ± 1.73 ^Cc^
Rl	41.35 ± 5.24 ^Bb^	30.28 ± 3.02 ^Ba^	17.98 ± 6.42 ^Bbc^	0.91 ± 5.31 ^Cc^	20.16 ± 3.28 ^Bb^	22.49 ± 1.92 ^Bb^
Rm	43.02 ± 6.73 ^Bab^	33.456 ± 6.57 ^Aab^	20.42 ± 5.12 ^Bb^	1.89 ± 3.83 ^ABab^	22.39 ± 3.31 ^Bb^	23.72 ± 2.14 ^Bb^
Rh	49.88 ± 4.04 ^ABa^	36.35 ± 4.33 ^Aa^	25.52 ± 4.53 ^Bb^	2.01 ± 2.57 ^ABab^	26.71 ± 2.27 ^ABa^	25.59 ± 3.71 ^Bb^
PC	49.52 ± 2.11 ^ABab^	36.17 ± 3.54 ^Aa^	25.95 ± 3.72 ^Bb^	2.04 ± 2.91 ^ABab^	26.09 ± 3.016 ^ABa^	25.88 ± 2.14 ^Bb^

N: normal control group. IM: immunosuppressive model. Rl: *R. mucilaginosa* ZTHY2 with low concentration. Rm: *R. mucilaginosa* ZTHY2 with medium concentration. Rh: *R. mucilaginosa* ZTHY2 with high concentration; PC: positive control. Different lowercase letters in the same column indicate significant difference (*p* < 0.05). Different capital letters in the same column indicate extremely significant difference (*p* < 0.01). The same or no letters indicate non-significant difference (*p* > 0.05).

**Table 5 animals-13-03376-t005:** Detection of T lymph subpopulations in mouse mesenteric lymph nodes.

Group	T Lymph Subpopulations				B Lymph Subpopulations	
	CD3^+^ (%)	CD4^+^ (%)	CD8^+^ (%)	CD4^+^/CD8^+^	CD19^+^ (%)	CD20^+^ (%)
N	48.31 ± 2.51 ^Aa^	27.02 ± 0.21 ^Aa^	25.91 ± 6.37 ^Aa^	4.03 ± 0.63 ^Aa^	30.71 ± 1.41 ^Aa^	33.04 ± 1.77 ^Aa^
IM	9.61 ± 2.83 ^Cc^	8.91 ± 1.21 ^Cc^	9.21 ± 1.04 ^Cc^	1.08 ± 0.94 ^Cc^	14.92 ± 1.21 ^Cc^	15.05 ± 0.97 ^Cc^
Rl	41.21 ± 1.41 ^Bb^	22.69 ± 1.22 ^ABa^	16.15 ± 1.72 ^Bb^	2.16 ± 1.17 ^bc^	21.97 ± 0.95 ^BCb^	26.71 ± 1.38 ^Bb^
Rm	43.26 ± 2.15 ^Abab^	24.65 ± 0.93 ^ABa^	17.03 ± 2.22 ^Bb^	2.94 ± 1.32 ^ABab^	23.18 ± 1.65 ^BCb^	27.82 ± 1.32 ^Bb^
Rh	46.84 ± 0.98 ^Aba^	26.65 ± 1.41 ^Aa^	18.91 ± 0.19 ^Bb^	3.01 ± 0.92 ^ABab^	27.18 ± 1.24 ^Bab^	29.53 ± 1.98 ^Bab^
PC	45.61 ± 1.61 ^Abab^	26.03 ± 0.94 ^Aa^	18.56 ± 1.42 ^Bb^	3.19 ± 1.52 ^ABab^	27.09 ± 1.32 ^Bab^	29.88 ± 0.99 ^Bab^

N: normal control group. IM: immunosuppressive model. Rl: *R. mucilaginosa* ZTHY2 with low concentration. Rm: *R. mucilaginosa* ZTHY2 with medium concentration. Rh: *R. mucilaginosa* ZTHY2 with high concentration; PC: positive control. Different lowercase letters in the same column indicate significant difference (*p* < 0.05). Different capital letters in the same column indicate extremely significant difference (*p* < 0.01). The same or no letters indicate non-significant difference (*p* > 0.05).

## Data Availability

The authors confirm that all the data used in the article supporting this study are available within the article.

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
