# Peer review of "Rhodotorula mucilaginosa ZTHY2 Attenuates Cyclophosphamide-Induced Immunosuppression in Mice"

_animals, 2023, doi:10.3390/ani13213376_

Round 1

Reviewer 1 Report

Comments and Suggestions for Authors

After reading the presented manuscript entitled "Rhodotorula mucilaginosa ZTHY2 attenuate cyclophosphamide-induced immunosuppression in mice" I have some comments:

1. The material/methods are not described in the Abstract.

2. The aim of the study is not presented in the Introduction.

3. Line 90, the section "Bacterial strain, antibodies and reagents" must be changed to "Microbial strains, ..." as Rhodotorula and Lactobacillus are mentioned in the paragraph.

4. Rhodotorula mucilaginosa can be a harmful microorganism. Please, add a paragraph for the pathogenic potential of the fungus and its role as a cause of severe infections in animals/humans, especially, if they are with a weakened immune system. 

Overall, the study is interesting, easy to read and understand.

Author Response

Responds to Reviewers’ Comments

Reviewer #1:

After reading the presented manuscript entitled "Rhodotorula mucilaginosa ZTHY2 attenuate cyclophosphamide-induced immunosuppression in mice" I have some comments:

1. Comment: The material/methods are not described in the Abstract.

Response: Thank you for your precious suggestion to improve our manuscript. We have added the main materials and methods in the abstract of the manuscript.

“Seventy-two male SPF mice were divided into six groups, C group (control), IM group (immunosuppressive model group) (+CTX), Rl, Rm and Rh groups (+CTX+low, middle and high concentration of R.mucilaginosa respectively) and PC (positive control) group (+CTX+LA). After 28 days feeding trial, blood samples were taken for biochemical and serum immunological analysis, thymus and spleen were collected to analyzed organ index, lymphocyte proliferation and differentiation, and antioxidant capacity. ”

2. Comment:The aim of the study is not presented in the Introduction.

Response: Thank you for your kind reminder. We have stated the purpose and significance of our research in the last paragraph of the introduction.

“In this study, in order to investigate the effect of R. Mucilaginosa ZTHY2 on immune protective and antioxidant function in CTX induced immunosuppressed mice, the mechanisms will be addressed by analyzing the immune organ index, blood biochemical, immunoglobulins, inflammatory factors, lymphocyte proliferation and differentiation. Our research provides a theoretical basis for the use of R. mucilaginosa ZTHY2 as immunopotentiator and attenuates the CTX-induced immunosuppression, and give an application perspective for explore in any other fields.”

3. Comment: Line 90, the section "Bacterial strain, antibodies and reagents" must be changed to "Microbial strains, ..." as Rhodotorula and Lactobacillus are mentioned in the paragraph.

Response: Thank you for your scientific review. We regret for our inappropriate use of “Bacterial”, and we have changed to Microbial strains.

4. Comment: Rhodotorula mucilaginosa can be a harmful microorganism. Please, add a paragraph for the pathogenic potential of the fungus and its role as a cause of severe infections in animals/humans, especially, if they are with a weakened immune system.

Response: Thank you for your advice. We have noticed that there were some reports of fungal cases in humans, such as skin infections, fungemia, or inflammation. However, the current reports are all secondary infections in patients was suffering frome tumor, Immunodeficiency disease, or chronic disease. And all reports mentioned that the infection of R. mucilaginosa is a rare case. There were more reports about development of R. mucilaginosa products. Therefore, in the beginning of the manuscript, we focused on the introduction of the prospect of R. mucilaginosa in food, industry and as a feed additive. We have modified the manuscript and added the content of the case report in the first paragraph of the introduction. The added part as below.

R. mucilaginosa possesses valuable biotechnological features, although there were a few cases reports of fungemia, inflammation, or skin infection in human, particularly in patients with immunodeficiency, tumors, or chronic disease [4-8].” Considering that the focus of the paper is on the immunomodulatory role of R. mucilaginosa, as well as the rarity of cases in animals. We didn't show this part as a separate paragraph.

Overall, the study is interesting, easy to read and understand.

We want to express our great appreciation to you  for your constructive comments and suggestions on our paper. We are looking forward to hearing from you soon. Best regards!

Reviewer 2 Report

Comments and Suggestions for Authors

The study is quite interesting and can provide support to improve the quality of life of severely immunosuppressed patients, however, I was surprised by a serious error. Rhodotorula mucilaginosa is not a bacterium, but a fungus, belonging to the phylum Basiciomycota, family basidiomycetes. It is not acceptable for authors to explore the biotechnological potential of a microorganism without at least knowing which kingdom it belongs to.

Additionally, I noticed some mistakes in the methodology, such as the solution of single cells from the spleen, for example, without mentioning the red blood cell lysis protocol for flow cytometry. Still regarding this methodology, the authors describe the use of the fluorochrome FITC coupled to two molecules, CD4 and CD19. It is necessary to also describe the fluorochrome coupled to the CD20 marker, otherwise the data obtained by flow cytometry are not reliable.

Regarding statistical analysis, what was the reason for choosing the Ducan post test?

I suggest a robust review of the work, including a brief presentation of the fungus with which the experiments were carried out.

Author Response

Reviewer #2

1. Comment: The study is quite interesting and can provide support to improve the quality of life of severely immunosuppressed patients, however, I was surprised by a serious error. Rhodotorula mucilaginosa is not a bacterium, but a fungus, belonging to the phylum Basiciomycota, family basidiomycetes. It is not acceptable for authors to explore the biotechnological potential of a microorganism without at least knowing which kingdom it belongs to.

Response: Thank you for your precious suggestion to improve our manuscript. We knew that Rhodotorula mucilaginosa is a fungus, and we defined it as “yeast” in the first sentence of the introduction. However, the language was not strict due to our careless wording using. We have checked and made changes in the text such as line 90: the section "Bacterial strain, antibodies and reagents" have changed to "Microbial strains”. Line 120: we change the “the concentration of bacterium was” to “the concentration was”.

2. Comment: Additionally, I noticed some mistakes in the methodology, such as the solution of single cells from the spleen, for example, without mentioning the red blood cell lysis protocol for flow cytometry. Still regarding this methodology, the authors describe the use of the fluorochrome FITC coupled to two molecules, CD4 and CD19. It is necessary to also describe the fluorochrome coupled to the CD20 marker, otherwise the data obtained by flow cytometry are not reliable.

Response: Thank you for your kind reminder. The red blood cell lysis protocol is mentioned in the “2.9 CCK-8 detection” and “2.10 Flow cytometry detection”, as show in the picture as below.

The fluorochrome coupled to the CD20 antibody have described in the “2. Materials and Methods” appeared in “2.1. antibodies and reagents” and “2.10. Flow cytometry detect the T and B cells species of spleen, thymus and mesenteric lymph nodes” respectively. The APC-CD20 antibody(Cat#:152107 ) was purchased from Biolegend.

3. Comment: Regarding statistical analysis, what was the reason for choosing the Ducan post test?

Response: Thank you for your kind reminder. We used one-way ANOVA to statistical analysis and followed by Duncan's to significance difference analysis among   multiple groups. Our expression is not clear enough, and we have made changes in the manuscript of Statistical analysis.  

4. Comment: I suggest a robust review of the work, including a brief presentation of the fungus with which the experiments were carried out.

Response: we really appreciate for your scientific review. The fungus of Rhodotorula mucilaginosa have mentioned in the first paragraph of the introduction. 

We want to express our great appreciation to you for your constructive comments and suggestions on our paper.  Best regards!

Reviewer 3 Report

Comments and Suggestions for Authors

The manuscript describes the immunoactivating effect of rhodoturula mucilaginos ZTHY2 in mice treated with cyclophosphamide. The work raises important issues because cyclophosphamide is used in cancer therapy and causes serious side effects in the form of leukopenia and blood morphology disorders. Therefore, finding an agent that could counteract these problems is of great importance for human therapy. Therefore, the manuscript appears to be aimed more at readers of medical journals devoted to human diseases.  In the introduction, the authors should present the use of cyclophosphamide in animals and its side effects and support it with appropriate literature to make it more suitably for publication in "Animals". The work also requires language correction and  the discussion  requires a thorough reconstruction. I suggest citing research on substances that have been used so far to counteract the side effects of cyclophosphamide or leukopenia. The choice of positive control also requires better justification - how well has the positive effect of lactobacillus on leukopenia been documented?

More specific comments:

line 17 I think that a better justification would be that companion animals are undergoing anticancer therapy and struggling with its side effects - immunosuppression

line 22 please rewrite: can enhance the immune and antioxidant function of the body

line 83 reference 29 lack a proper citation

line 102(six to nine weeks) - it is unclear

line 111 bacterium or yeast?

line 130 there are no results for RBC, HGB, PLT

line 162 2.9 please rewrite the title of subchapter

line 182 statistical analysis: Duncan's is a post hoc test what was the main test : ANOVA? were the data normally distributed?

line 198 please rewrite the sentence: the body weight was increased not the goup

  Table 1 caption please add the description of the letters

line 228 was _ were

lines 437-438 Please rewrite the sentence

line 460 metabolic response? please rewrite the senetence

line 502 please provide the references

Comments on the Quality of English Language

.

Author Response

Reviewer # 3

1. Comment: The manuscript describes the immunoactivating effect of rhodoturula mucilaginos ZTHY2 in mice treated with cyclophosphamide. The work raises important issues because cyclophosphamide is used in cancer therapy and causes serious side effects in the form of leukopenia and blood morphology disorders. Therefore, finding an agent that could counteract these problems is of great importance for human therapy. Therefore, the manuscript appears to be aimed more at readers of medical journals devoted to human diseases. In the introduction, the authors should present the use of cyclophosphamide in animals and its side effects and support it with appropriate literature to make it more suitably for publication in "Animals". The work also requires language correction and the discussion requires a thorough reconstruction. I suggest citing research on substances that have been used so far to counteract the side effects of cyclophosphamide or leukopenia. The choice of positive control also requires better justification - how well has the positive effect of lactobacillus on leukopenia been documented?

Response: Thank you for your scientific review. The primary application of cyclophosphamide in animals pertains to its utilization as an antineoplastic agent in companion animals and cyclophosphamide is routinely used to prepared animal immunosuppressive models, and there are a lot of papers to use cyclophosphamide to prepare kinds of animal models, that is why we used the cyclophosphamide in our immunosuppressive models. We have presented the use of cyclophosphamide in animals and its side effects in the introduction. The modified part as below: “CTX is also commonly used as a cancer treatment in companion animals [27-30], and the administration of CTX often leads to adverse effects in treated companion animals such as lethargy, moderate alopecia, vomiting, anorexia, anemia, and haematuria [31-33]. In addition, CTX is routinely used to induce immunosuppression in animal models. [34-37].”

We presented strategies to reduce the side effects of cyclophosphamide, and pointed out the advantages of R. mucilaginosa in the first paragraph of the discussion part. “At present, there are a lot of studies on exploring immunomodulators, such as peptides from soybean, nucleic acid combine with microRNA, extracts of plants and medicinal herbs, peptides from aquatic organism, and etc.[45-49]. However, R. mucilaginosa ZTHY2 has a great advantage that it is widely available in the ocean, easy to isolate and culture, and the culture cost is low, hence, it has a good application prospect and is worth further research. ”

In the discussion part we gave the justification for choice of lactobacillus as positive control. The modified part as below: “The activity of LA in stimulating the immune system has been verified in a variety of species. The immunomodulatory effects of LA were demonstrated in mice as early as 1986[50]. Subsequently, it has been confirmed in a variety of animals such as laying hens, broiler chickens, zebrafish, and weaned calves [51-54]. And different strains of LA ameliorated the CTX-induced immunosuppression in Mice, and the protect function of LA frome CTX was reflected in that LA reversed the body and spleen weights, blood RBC and WBC levels, and splenocyte and bone marrow cells, increased T cell proliferation, induced the expressions of IL-2 and IFN-gamma in T cells, restored the phagocytosis of macrophages in mice and cytotoxicity of NK and cytotoxic T cells [55-58]. So, we used LA as a positive control in our experiment.”

2. Comment: line 17 I think that a better justification would be that companion animals are undergoing anticancer therapy and struggling with its side effects – immunosuppression

Response: Thank you for your precious suggestion to improve our manuscript. Cyclophosphamide (CTX) is an antineoplastic drug for lymphoma and an immunosuppressant to treat autoimmune diseases, such as severe systemic lupus erythematosus. Rhodotorula mucilaginosa attenuate the side effects of CTX and has clinical potential as a protective agent for CTX in cancer treatment. As you said, companion animals are playing an important role in our lives, and companion animals suffer from the side effects of CTX during cancer treatment. Rhodotorula mucilaginosa exhibits promising potential in effects of CTX in companion animal cancer therapy. Meanwhile, CTX is also routinely used to induce immunosuppression in animal models. Immunosuppression is common in livestock production. We are currently assessing the immune system enhancing effects of Rhodotorula mucilaginosa in waterfowl. Therefore, due to the systematic arrangement of our work, we have primarily emphasized the role of Rhodotorula mucilaginosa in attenuating CTX-induced immunosuppression in the introduction and abstract of the manuscript. Thank you for your precious advice we also added the introduction for CTX applying in companion animals.

3. Comment: line 22 please rewrite: can enhance the immune and antioxidant function of the body

Response: Thank you for your advice. We have revised the sentence as suggested.

4. Comment: line 83 reference 29 lack a proper citation

Response: Thank you for your kind reminder. We have added the information of reference 29.

5. Comment: line 102(six to nine weeks) - it is unclear

Response: Thank you for your comment. The “six to nine weeks” was used to describe the age of the mice during the experiment, but our expression is not precise enough. We have revised it to “six to nine weeks old”

6. Comment: line 111 bacterium or yeast?

Response: Thank you for your kind reminder. The Rhodoturula mucilaginos ZTHY2 is yeast, we made a mistake in our presentation, and we have corrected two errors in the manuscript line 111, and line 90.

7. Comment: line 130 there are no results for RBC, HGB, PLT

Response: Thank you for your kind reminder. We had tested the RBC, HGB, PLT, but in the final manuscript, we deleted these results to simplify the experimental results, since those are not closely related to the main content of the manuscript. So we have deleted “RBC, HGB, PLT ”in line 130.

8. Comment: line 162 2.9 please rewrite the title of subchapter

Response: Thank you for your comment. The title has been rewrite as “Proliferation of T and B lymphocytes in spleen and thymus in vitro.

9. Comment: line 182 statistical analysis: Duncan's is a post hoc test what was the main test : ANOVA? were the data normally distributed?

Response: Thank you for your kind reminder. We used one-way ANOVA to statistical analysis and followed by Duncan's to significance difference analysis among   multiple groups. Our expression is not clear enough, and we have modified in the manuscript of statistical analysis.

10. Comment: line 198 please rewrite the sentence: the body weight was increased not the goup

Response: thank you for your scientific review. We have rewritten the sentence as “the body weight of the R groups and PC group were significantly increased”.

11. Comment: Table 1 caption please add the description of the letters

Response: Thank you for your comment. We don't understand what you mean. In the manuscript, we have described for all the abbreviation and letters that appear in the table. The caption for table 1 was “N: normal control group. IM: immunosuppressive model. Rl: R. mucilaginosa ZTHY2 with low concentration. Rm: R. mucilaginosa ZTHY2 with middle concentration. Rh: R. mucilaginosa ZTHY2 with high concentration. PC: positive control. Different lowercase letters in the same column indicate significant differences (P<0.05). Different capital letters in the same column indicate extremely significant difference (P<0.01). The same or no letters indicate non-significant differences (P>0.05).” We also checked the captions of all tables.

12. Comment: line 228 was _ were

Response: Thank you for your kind reminder. We have modified the word.

13. Comment: lines 437-438 Please rewrite the sentence

Response: Thank you for your kind reminder. The sentence was modified as: “R. mucilaginosa ZTHY2 exhibits immunostimulatory properties and regulates the gastrointestinal microbiome. Furthermore, our study confirmed its ability to enhance immune response and antioxidant capacity in mice under immunosuppressive conditions”.

14. Comment: line 460 metabolic response? please rewrite the senetence

Response: Thank you for your kind reminder. The sentence was modified as: “The blood actively engages in metabolic reactions and plays a crucial role in maintaining homeostasis ”

15. Comment: line 502 please provide the references

Response: Thank you for your precious suggestion. In the manuscript we said “our study was consistent with previous studies”, the “previous studies” means to reference 56. We modified the sentence in order to make the expression more accurate.

 “Study have showed that when animals suffered from CTX, the secretion levels of CD3+, CD4+, CD8+, and CD4+/CD8+ will be reduced, resulting in imbalance of Th1/Th2 cell proportion, decreased immune function, and induced secondary infection[56], the findings of our study consistent with this conclusion. The results demonstrated a significant decrease in the levels of immune cytokines in the serum of mice in the IM group,The results of this study showed that the content of immune cytokines in serum of mice in the IM group was significantly decreased, as well asand the secretion of CD3+, CD4+, CD8+, and CD4+/CD8+ in spleen, thymus and mesenteric lymph nodes was inhibited, it was consistent with previous studies.”

We want to express our great appreciation to you for your constructive comments and suggestions on our paper.  Best regards!

Yours sincerely,

Kai Kang

College of Coastal Agriculture, Guangdong Ocean University

Zhanjiang, Guangdong 524088, China.

E-mail address: kangkai610@126.com

Round 2

Reviewer 1 Report

Comments and Suggestions for Authors

Thank you for your answers. After the corrections, the manuscript significantly improved. 

Author Response

Thank you for all your kind comments.

Reviewer 3 Report

Comments and Suggestions for Authors

The manuscript has been improved, however I still have two questions:

1) Were the data normally distributed/ it should be checked if  ANOVA is used for statistical manalysis

2) the animals were in different age (6-9 weeks of age) Did you check if the effect of age is significant? Are you sure it does not interfer with the results?

Style should be improved:

for example line 476 . So, we used...

Author Response

Thank you very much for taking the time to review this manuscript. Please find the detailed responses below and the corresponding revision in track changes in the re-submitted files.

Point-by-point response to Comments and Suggestions for Authors

Comments 1: Were the data normally distributed/ it should be checked if  ANOVA is used for statistical manalysis

Response 1: Thank you for pointing this out. The data approximately follows a normal distribution, with most groups of data having small standard deviation, which further indicates the presence of a normal distribution. However, due to the limited number of replicates in each group, A few groups’ data do not exemplify a typical normal distribution.

Comments 2: the animals were in different age (6-9 weeks of age) Did you check if the effect of age is significant? Are you sure it does not interfer with the results?

Response 2: Thank you for your precious suggestion to improve our manuscript. The age of the animals in the experiment lasted from 6 to 9 weeks, as the entire duration of the experiment spanned four weeks. Our animals were all at the same age before the experiment began. The focus of this paper is not focus on whether R.mucilaginosa has different effects on animals of different ages. I think you have raised a new and interesting issue that we will consider in subsequent experiments.

Response to Comments on the Quality of English Language

Style should be improved:

for example line 476 . So, we used...

ResponseThank you for your kind reminder. We have carefully revised the language and grammar throughout the text. We have made several modification in the text

Line 476:”So, we used LA as a positive control in our experiment” modified to “So, the LA was used as a positive control in our experiment.”

Line 125:”having six mice/replicate.” modified to “with six mice in each replicate.”

Line 170: “respectivly” modified to “respectively”

Line 217: “ the R groups and PC group was” modified to “the R groups and PC group were”

Line 265, 298: “was” modified to “were”

Line 315 : delete ”which was ”

Line 344:” CD20+” modified to “ CD20+”
